# Safety and Efficacy of the Nit-Occlud^®^ Coil for Percutaneous Closure of Various Sizes of PDA

**DOI:** 10.3390/jcm11092469

**Published:** 2022-04-28

**Authors:** Seyong Jung, Jaehee Seol, Jaeyoung Choi, Keesoo Ha

**Affiliations:** 1Division of Pediatric Cardiology, College of Medicine, Yonsei University, Seoul 03722, Korea; jung811111@yuhs.ac; 2Department of Pediatrics, Yonsei University Wonju College of Medicine, Wonju 26426, Korea; seoljh0623@gmail.com; 3Department of Pediatrics, College of Medicine, Korea University, Seoul 02841, Korea

**Keywords:** patent ductus arteriosus, therapeutic occlusion, procedure

## Abstract

Most interventionalists use the Amplatzer Duct Occluder (ADO) or the Nit-Occlud^®^ Coils (NOC) to close patent ductus arteriosus (PDA). Data regarding the success and effect of NOCs in the occlusion of large PDAs are insufficient. We aimed to investigate whether the PDA occlusion of large PDAs using NOC is safe and efficient for all ages. This was a retrospective study involving 361 pediatric and adult patients who underwent the transcatheter closure of PDA using NOC over the past 21 years for all PDA sizes and ages. The sizes of PDA were classified as small, moderate, and large. A comparison of the aortic pressure before and after PDA occlusion using NOC showed significant differences in terms of systolic and pulse pressures for all age groups (*p* < 0.05). The rate of the residual shunts of NOC was 2%, while the rate of complete occlusions of NOC was 98% at 12 months after occlusion regardless of the shape of PDA. The complication rate with PDA occlusion using NOC was 5%. PDA occlusion using NOC is as effective and safe as ADO for the occlusion of PDA of all sizes. Therefore, PDA occlusion using NOC can be a safe and feasible procedure to close various sizes and types of PDA without complications.

## 1. Introduction

Device procedures for the closure of patent ductus arteriosus (PDA) have increasingly been preferred over open-heart surgery due to their associated lower mortality and morbidity; however, their costs are more expensive than open surgery [1]. A number of techniques and occluders, including coils and devices, have been developed for the transcatheter closure of PDA over recent decades ever since the percutaneous closure of PDA for the first time in 1967 by Porstmann [2]. 

There is currently no consensus on which types of devices are more suitable for the different sizes of PDA. Nevertheless, data have shown that the Amplatzer ductal occluder (ADO) device and Nit-Occlud^®^ coil (NOC, PFM Medical AG, Cologne, Germany) are more efficient and safer for a relatively large PDA [3]. The transcatheter closure of PDA using the NOC is a safe and simple method, which can be performed for all age groups. However, most investigations on the NOC have been limited to small or moderate PDAs [4,5,6,7]. Data regarding the safety and efficacy of NOC in the occlusion of large PDAs have not been sufficiently reported compared to those of ADO.

In this study, we aimed to determine the safety and efficacy of NOC in the transcatheter closure of large PDAs for all ages, including pediatric and adult patients.

## 2. Materials and Methods

### 2.1. Patients

We retrospectively investigated PDA patients who underwent PDA occlusion using NOC between April 1995 and November 2016 (21 years) at Severance Cardiovascular Hospital, Yonsei University. A total of 361 patients, including pediatric and adult patients, were enrolled in this study. 

The enrolled patients were divided into five pediatric age groups and one adult age group: 6 months of age to <1 year of age, 1 year of age to <6 years of age, 6 years of age to <12 years of age, 12 years of age to <18 years of age, and 18 years of age and above. The sizes of PDA were classified as small, moderate, and large, according to the scales of 2 and 4 mm (three groups).

We retrospectively analyzed medical data, echocardiographic results, and angiographic findings with hemodynamic characteristics. The demographics, the sizes and types of PDA, and the size relationships between PDA and NOC were analyzed. Furthermore, the regional sizes of PDA, such as aortic ampulla (AoA), pulmonary ampulla (PuA), length, and PDA isthmus (PI), and the regional sizes of NOC, such as the distal diameter (DD) and proximal diameter (PD), were also analyzed. The size gaps between the PDA and used NOC reflect the relationship between the regional sizes of PDA and NOC; these include size gaps between the AoA of PDA and DD of NOC, and those between PuA and PD of NOC, which are anatomically related to each other. Moreover, since the size of PI of PDA is the standard for NOC selection, it is important to know the size differences (gaps) between PI and DD, and between PI and PD. Echocardiography was performed to assess the position of the NOC, the presence of residual shunts around PDA, and major or minor complications. Follow-up echocardiography analyses were conducted at the immediate time of the procedure, the next day, and at 1 month, 6 months, and 1 year after the procedures.

Hemodynamic characteristics in the angiographic findings were obtained by observing the changes in blood pressure and the ratio of pulmonary blood flow to systemic blood flow (Qp/Qs) before and after PDA occlusion using NOC during cardiac catheterization. The aortograms were obtained in the right anterior oblique angle (30°) and lateral angle (90°) to confirm the positions of PDA and NOC. Additionally, the maximum diameters of the narrowest portion of the PDA (PDA isthmus, PI) were measured on frozen images. Furthermore, balloon-occlusive diameters of PDA were measured to calculate the accurate size of PDA, if needed. The residual shunts and configurations were identified within 10 to 15 min via an aortic aortogram after the implantation of NOC.

Mild narrowing (peak velocity < 1.5 m/s) of the pulmonary artery (PA), mild narrowing of the aorta (peak velocity < 1.5 m/s), and transient weakness of the arterial pulse were regarded as minor complications. A peak velocity of >1.5 m/s was considered as stenosis, which indicated the presence of a major complication. The procedural success in the PDA occlusion using NOC was considered as conditions with the absence of a residual shunt, stenotic lesions of the aorta, PA, and any other serious complications.

### 2.2. Ethics Statement

This study was approved by the Institutional Review Board of the Severance Hospital, Yonsei University Health System (4-2021-0800). The requirement for informed consent was waived due to the retrospective nature of the study.

### 2.3. Statistical Analysis

All data were represented as mean ± standard deviation, and a *p* value < 0.05 was regarded as statistically significant. Statistical analyses were performed using the SPSS 20 software (IBM Corp., Armonk, NY, USA). Additionally, the values were compared among each group using a non-parametric test (Mann–Whitney U test, χ2 test, and Wilcoxon test).

## 3. Results

The mean age and body weight of all patients, including pediatric and adult patients, were 80 months (6.7 years) and 20 kg (not shown in Table 1), respectively. The pressure differences between the pediatric and adult groups were significant in terms of the following: systole, mean, and pulse pressures in the aorta before PDA occlusion using NOC (*p* < 0.01), pulse pressure in the main PA before PDA occlusion using NOC (*p* < 0.05), systole, diastole, and mean pressure gaps between the aorta and the main PA before PDA occlusion using NOC (*p* < 0.01), and systole and pulse pressure in the aorta after PDA occlusion using NOC (*p* < 0.05) (Table 1).

The size gaps between PDA and NOC showed significant differences in each pediatric and adult group. The size gaps between AoA and DD among the pediatric groups showed significant differences (*p* = 0.001), and their mean size gap was approximately 2 mm (1.7 ± 3.4 mm). The size gaps between PuA and PD, PI and DD, and PI and PD among the pediatric groups did not show significant differences, and their mean size gaps were approximately 0 mm (−0.3), −4 mm (−3.5), and −3 mm (−2.6), respectively. The size gaps of AoA and DD, PI and DD, and PI and PD between the pediatric and adult groups showed significant differences (*p* < 0.001, respectively). Additionally, their mean size gaps in the adult group were approximately 5 mm (5.1), −5 mm (−5.0), and −2 mm (−1.8), respectively. The size gaps of PuA and PD between the pediatric and adult groups did not show significant differences, and the mean size gap in the adult group was observed to be approximately 1 mm (1.3). The Qp/Qs before PDA occlusion using NOC showed significant differences not only between the pediatric and adult groups (*p* = 0.001), but also among the pediatric groups (*p* = 0.023). However, the Qp/Qs after the PDA occlusion using NOC did not show statistical differences among all age groups (Table 1; Figure 1).

The most common size group of PDA was the moderate size. Additionally, age and body weight showed significant differences among the size groups of PDA (*p* < 0.001). The pressures among the size groups of PDA showed significant differences as follows: systolic and pulse pressure of the aorta (*p* < 0.001); systolic, diastolic, mean pressure of the main PA (*p* < 0.05); and systole pressure gaps between the aorta and main PA (*p* < 0.001) before PDA occlusion using NOC. However, the systole, diastole, mean, and pulse pressures of the aorta after PDA occlusion using NOC did not show significant differences among the PDA size groups. The size gaps between PDA (AoA, PuA and PI) and NOC (DD and PD) showed significant differences among the size groups of PDAs; the size gap between AoA and DD and the size gap between PI and PD showed significant differences (*p* < 0.001, respectively). The Qp/Qs before PDA occlusion using NOC among the different PDA size groups and Qp/Qs between before and after PDA occlusion using NOC showed significant differences (*p* < 0.001, respectively). The rate of residual shunts of NOC decreased from 25% to 2%, while the rate of complete occlusions of NOC increased from 75% to 98% from the immediate time of the procedure to 12 months after PDA occlusion using NOC (Table 2).

The frequency order of the PDA types in the patients was as follows: A (conical) > E (elongated) > B (window) > C (tubular) > D (complex). Additionally, the one in the pediatric group was the same as those in all patients. The frequency order of PDA types in the adult group was A > B > E = C > D, although the number of adult patients was small. The frequency of type E (elongated) PDA in the pediatric group was higher than that in the adult group (*p* = 0.048) (Table 3). 

A comparison of the systole and the pulse pressure of the aorta before and after PDA occlusion using NOC showed significant differences in the total age group and pediatric age group (*p* < 0.05). A comparison of the Qp/Qs values before and after PDA occlusion using NOC showed significant differences in all age groups (*p* < 0.001, respectively) (Table 4). 

The most common diagnosis of congenital heart diseases (CHDs) accompanying PDA occluded by NOC was the simple PDA without other CHDs. Other common diagnoses of CHDs accompanying PDA occluded by NOC were the secundum atrial septal defect, ventricular septal defect, endocardial cushion defect, and the Taussig–Bing anomaly (Table 5).

The most common complication associated with PDA occlusion using NOC was a residual shunt, followed by disturbance in the flow of PA by NOC. The total proportion of complications associated with PDA occlusion using NOC was 5% (Table 6).

## 4. Discussion

The pressures prior to PDA occlusion using NOC showed statistical differences not only between pediatric and adult groups, but also among different pediatric age groups. However, these differences occur only due to the gradual physiological increase in normal blood pressures according to the age and body surface area [8]. Moreover, the gradual increase in the aortic ampulla, pulmonary ampulla, length, and isthmus of PDA prior to the occlusion of NOC with age occurs due to the same reasons.

The occlusion of PDA has an influence on the systemic blood pressure and cerebral circulation of pediatric patients. Furthermore, it leads to an increase in the systemic systolic, diastolic, and mean pressures after the percutaneous closure of PDA [9]. The ratio of Qp/Qs after PDA occlusion using a coil showed a dramatic decrease due to the decrease in extra pulmonary blood flow compared to that before the coil occlusion of PDA [10]. These facts are consistent with our investigation and are well-known physiological characteristics observed after PDA occlusion using a coil. It was meaningful that the pressure changes before and after PDA occlusion using NOC were established basically according to chronological ages. 

According to the instruction manual of NOC on the PFM medical AG website (Nit-Occlud^®^ PDA. Available Online: https://www.pfmmedical.com/productcatalogue/occluder/nit_occludr_pda/index.html (assessed on 15 March 2022)), the diameter of NOC should be selected as follows: the distal diameter of the coil (DD) should be a maximum of 2 mm larger than that of the aortic ampulla (AoA, represented as D2 in the instructions). The DD should be minimal and should be 3 to 4 mm larger than the PDA isthmus (PI, represented as D1 in the instructions). However, these instructions could be confusing or unperceivable; therefore, we created the following equations for the sizes of the PDA and NOC used based on Table 1:PuA (=PI + 3) = PD (=PI + 3) > PI < DD (= PI + 4) < AoA (=PI + 6) in children (the unit mm)
PuA (=PI + 3) > PD (=PI + 2) > PI < DD (= PI + 5) < AoA (=PI + 10) in adults (the unit mm)

These novel formulae are feasible and understandable and can definitely help in selecting the NOC sizes in a stable manner according to the various sizes and morphologies of PDA. The size relationships between the regions of PDA and NOC are illustrated in Figure 2 and Figure 3.

Almost all previous studies reported that NOC was effective and safe for small to moderate PDAs [4,5]. Our result showed that 50 patients had large PDA, with a size ranging from 4 to 9 mm in 27 pediatric (age ≤ 6 months; 1, 7–12 months; 3, 1–6 years; 13, 7–12 years; 7, 12–18 years; 3) and 23 adult patients. The large PDA accounted for the largest proportion in the adult group, and it was the most common PDA size at the age of 1–6 years compared to all other pediatric age groups. The rate of residual shunts in the large PDA group decreased to 6% at 12 months from 26% immediately after PDA occlusion using NOC. Furthermore, the success rate in the large PDA group was 94% in 12 months after PDA occlusion using NOC. Three patients in the large PDA group had residual shunts. Among the three, one had a trivial residual shunt, while two had moderate residual shunts, which were occluded by additional NOCs. The rate of persistent residual shunts was higher in the large-sized PDA group than those in small- or moderate-sized PDA groups. However, the rate of the residual shunt in large PDA was 6%, which could be controlled and treated with additional coils or secondary devices. 

The complications of patients with large PDA occluded by NOCs included having two residual shunts and atrial fibrillation. The two residual shunts were treated via the implantation of extra coils, while atrial fibrillation was treated with medication. Therefore, the PDA occlusion using NOC is an effective and safe procedure for the occlusion of large PDAs in addition to small to moderate PDAs, without major complications. 

The geometric shapes of PDA devices were developed since angiographic classification of PDA types was established by Krichenko A et al. [11]. Some studies reported that the frequency order of PDA types, which consisted of mostly pediatric patients, was as follows: conical > elongated > tubular > window > complex or conical > tubular > elongated > window > complex [12,13]. Most studies reported that the conical type was the most common type of PDA, while the rarest one was the complex type. The frequency order of PDA types among our study patients, consisting mostly of children and a few adults, was as follows: conical > elongated > window > tubular > complex. Furthermore, the frequencies of the elongated and window types in our study were relatively higher compared with those in other studies. The frequencies of the elongated and window types in the pediatric group of our study were relatively high compared with those in the adult group of our study. The ratio of the window-type PDA was evenly distributed among small, moderate, and large PDA groups (6/101 = 6%, 16/210 = 8%, and 4/50 = 8%, respectively). The elongated-type PDA, in addition to the fetal (F)-type PDA, was relatively common in pre-term infants [14]. Our study shows that the smaller the size of the PDA and the younger the age of the patient, the higher the ratio of elongated type PDA is (small PDA group 34/101 = 34%, moderate PDA group 24/210 = 11%, and large PDA group 4/50 = 8%) (*p* < 0.001, small vs. moderate, small vs. large PDA, respectively). The PDA occlusion using NOC did not show significant differences in the success rate according to the types of PDA, and it showed high success rates evenly in each group classified according to the PDA type. Therefore, the PDA occlusion using NOC is an effective procedure regardless of the morphological classification of PDA.

One of the characteristics of PDA hemodynamics is having a peripheral bounding pulse and wide pulse pressure in PDA patients [15]. The pulse pressure and pulmonary blood flow before and after PDA occlusion using NOC in our study showed significant decreases in the entire patient group and pediatric patient group. Moreover, this fact reflects the hemodynamic characteristics according to the presence or absence of a PDA. It has been reported that the diastolic pressure after PDA occlusion using devices increased [10]. However, our results did not show significant increases after PDA occlusion using NOC. 

Most studies reported that there was neither mortality in any group nor serious complications associated with PDA occlusion using NOC [16,17]. The main complications associated with NOC intervention at 12 months after the procedure, in this study, were the presence of residual shunts and flow disturbances of PA. The rate of total complications associated with PDA occlusion using NOC was as low as 5%, while the degree of complication in the large PDA group was very low at 1%. Therefore, PDA occlusion using NOC is a safe and feasible procedure in occluding large PDAs without resulting in major complications.

### Limitations

Pressure changes in the pulmonary artery before and after PDA occlusion using NOC were not compared because of insufficient main PA pressure data after PDA occlusion using NOC. Nevertheless, the percutaneous closure of PDA decreases immediately and continues to decrease gradually [18].

## 5. Conclusions

In summary, the novel formulae of the size relationships between the regions of PDA and NOC are feasible and understandable. Furthermore, they can be used to select NOC sizes in a stable manner, according to the various sizes and morphologies of PDA. PDA occlusion using NOC is as effective and safe as ADO for the occlusion of large PDAs in addition to small and moderate PDAs. Moreover, PDA occlusion using NOC is a feasible procedure, regardless of the morphological classification of PDA. Therefore, PDA occlusion using NOC is a safe and effective procedure for occluding various sizes and types of PDA without prominent complications. 

## Figures and Tables

**Figure 1 jcm-11-02469-f001:**
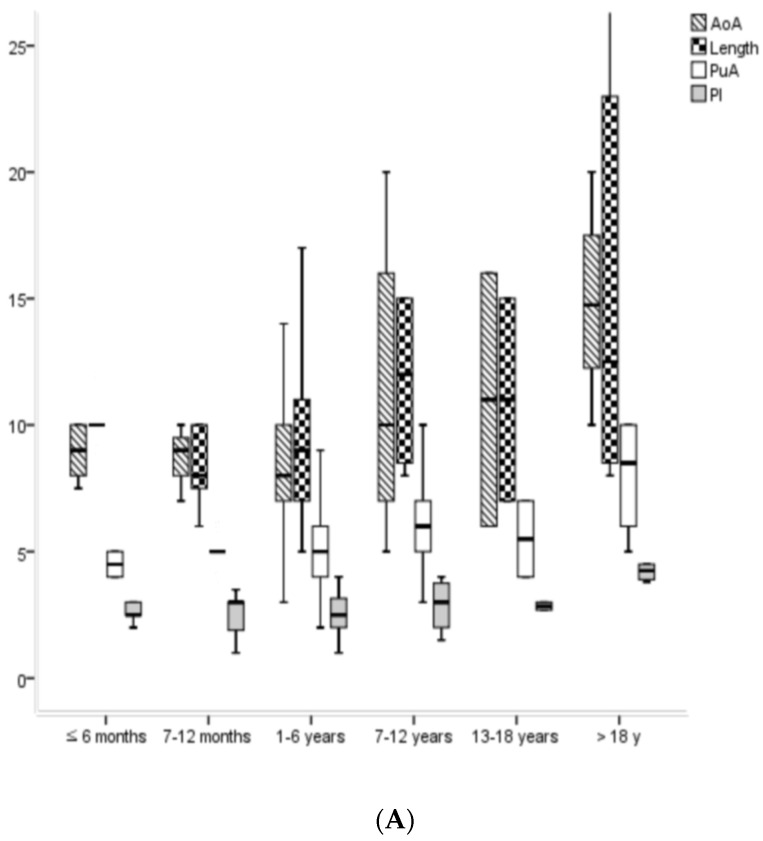
The changes in PDA sizes (**A**) and size gaps between PDA and NOC (**B**) according the chronological age groups. Ao, aorta; AoA, aortic ampulla of PDA; NOC, Nit-Occlud^®^ coil; PDA, patent ductus arteriosus; PuA, pulmonary ampulla of PDA; PI, PDA isthmus; DD, distal diameter of NOC; PD, proximal diameter of NOC.

**Figure 2 jcm-11-02469-f002:**
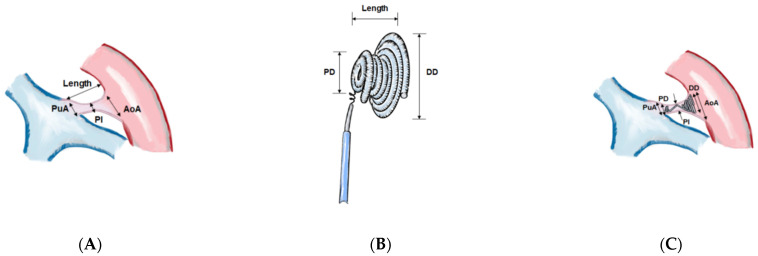
Illustration of the size relationships between regions of PDA and Nit-Occlud^®^ Coils (NOC). The schematic illustration of (**A**) the regions of PDA, (**B**) the regions of NOC, and (**C**) the installed coil inside PDA. PuA; pulmonary ampulla of PDA, AoA; aortic ampulla of PDA, PI; PDA isthmus, PDA; patent ductus arteriosus, PD; proximal diameter of NOC, NOC; Nit-Occlud^®^ coil, DD; distal diameter of NOC.

**Figure 3 jcm-11-02469-f003:**
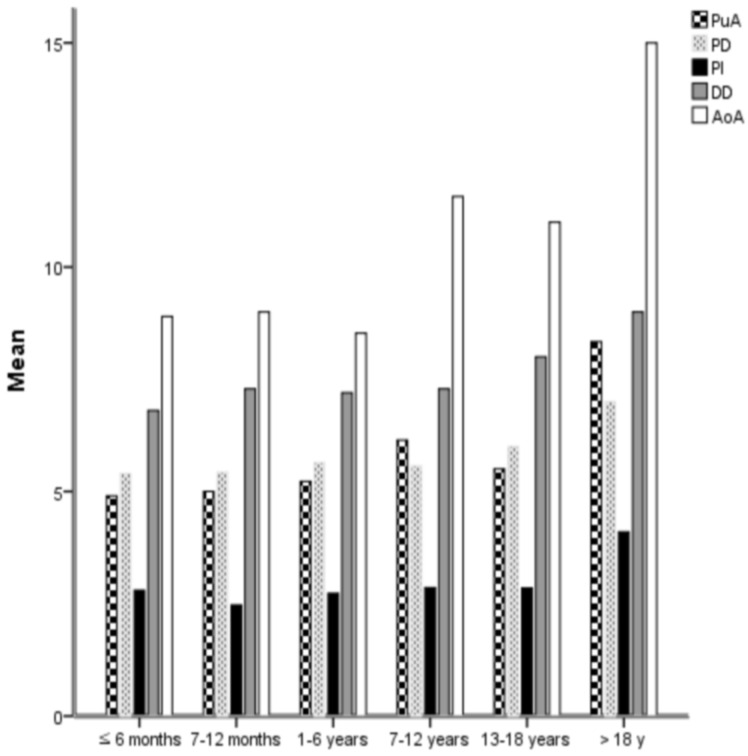
The size relationships between the regions of PDA and NOC; PI of PDA with other regions of PDA and NOC. It is useful to select the size of NOC by measuring the sizes of PI, PuA, and Ao according to chronological age. Ao; aorta, AoA; aortic ampulla of PDA, NOC; Nit-Occlud^®^ coil, PDA; patent ductus arteriosus, PuA; pulmonary ampulla of PDA, PI; PDA isthmus, DD; distal diameter of NOC, PD; proximal diameter of NOC.

**Table 1 jcm-11-02469-t001:** Characteristics of subjects with PDA occluded by NOC according to the chronological age groups.

	Pediatrics (0–18 y)	Age ≤ 6 Months	Age 7–12 Months	Age 1–6 Years	Age 7–12 Years	Age 13–18 Years	*p* Value (Among Pediatrics)	Adults (>18 y)
Number	325	26	57	175	57	10		36
Age (months)	41 ± 40	4 ± 2	10 ± 2	34 ± 18	103 ± 23	178 ± 21	<0.001	451 ± 185
Body weight (kg)	16 ± 11	7.0 ± 2.6	9.0 ± 1.6	14 ± 5	30 ± 10	56 ± 13	<0.001	57 ± 9
Pre-Ao Pr (mmHg)								
Systole	111 ± 17	100 ± 10	102 ± 15	112 ± 16	125 ± 21	121 ± 13	<0.001	129 ± 26
Diastole	63 ± 14	54 ± 10	54 ± 10	64 ± 13	76 ± 15	76 ± 10	<0.001	66 ± 11
Mean	83 ± 15	72 ± 10	74 ± 13	84 ± 14	96 ± 16	93 ± 15	<0.001	89 ± 11
Pulse Pr	50 ± 15	48 ± 10	49 ± 13	48 ± 10	49 ± 12	45 ± 7	0.748	67 ± 14
Pre-MPA Pr								
Systole	29 ± 9	29 ± 9	26 ± 6	29 ± 7	34 ± 13	36 ± 19	0.001	30 ± 8
Diastole	12 ± 6	11 ± 6	11 ± 5	12 ± 5	15 ± 8	15 ± 12	0.012	11 ± 4
Mean	19 ± 7	18 ± 6	18 ± 5	19 ± 6	23 ± 11	24 ± 16	0.003	19 ± 6
Pulse Pr	17 ± 6	17 ± 7	15 ± 5	16 ± 5	19 ± 7	21 ± 10	0.002	19 ± 5
Pr gaps of Pre-Ao and pre-MPA								
Systole	82 ± 17	71 ± 15	76 ± 14	83 ± 16	95 ± 15	84 ± 23	<0.001	100 ± 22
Diastole	50 ± 14	42 ± 13	43 ± 11	51 ± 13	64 ± 12	60 ± 11	<0.001	55 ± 11
Mean	63 ± 15	53 ± 14	56 ± 13	64 ± 14	75 ± 11	68 ± 18	<0.001	71 ± 10
Post-Ao Pr								
Systole	114 ± 15	95 ± 9	113 ± 6	115 ± 15	112 ± 19	123 ± 15	0.027	140 ± 17
Diastole	67 ± 12	47 ± 10	63 ± 6	68 ± 10	72 ± 24	73 ± 10	0.027	73 ± 3
Mean	86 ± 11	67 ± 6	83 ± 3	87 ± 10	87 ± 19	93 ± 16	0.035	90 ± 10
Pulse Pr	48 ± 14	47 ± 13	50 ± 10	47 ± 13	40 ± 9	50 ± 18	0.874	64 ± 25
PDA sizes (mm)								
AoA	7.9 ± 3.6	7.3 ± 1.7	7.4 ± 2.6	7.6 ± 3.4	9.8 ± 5.5	10.5 ± 4.9	0.003	14.7 ± 5.2
PuA	5.3 ± 1.7	4.9 ± 1.2	5.0 ± 0.6	5.3 ± 1.8	6.4 ± 2.3	5.5 ± 2.1	0.061	8.0 ± 2.4
Length	9.2 ± 3.5	8.3 ± 2.8	8.7 ± 3.2	8.9 ± 3.1	11.9 ± 5.1	12.1 ± 3.4	<0.001	14.8 ± 5.9
PI	2.3 ± 1.0	2.3 ± 1.2	2.1 ± 0.9	2.3 ± 0.9	2.6 ± 1.2	2.9 ± 0.8	0.023	4.4 ± 1.6
Size gaps of PDA and NOC (mm)								
AoA—DD	1.7 ± 3.4	0.8 ± 3.5	1.7 ± 2.1	1.3 ± 3.1	3.9 ± 5.2	2.8 ± 4.9	0.001	5.1 ± 5.8
PuA—PD	−0.3 ± 1.6	−0.5 ± 1.2	−0.4 ± 0.8	−0.4 ± 1.8	0.6 ± 1.5	−0.5 ± 2.1	0.707	1.3 ± 2.5
PI—DD	−3.5 ± 1.4	−3.6 ± 1.2	−3.4 ± 1.2	−3.6 ± 1.4	−3.2 ± 1.6	−4.4 ± 1.8	0.192	−5.0 ± 2.4
PI—PD	−2.6 ± 1.0	−2.7 ± 1.0	−2.6 ± 0.9	−2.6 ± 1.0	−2.2 ± 1.4	−2.6 ± 0.8	0.365	−1.8 ± 1.9
Pre-Qp/Qs	1.4 ± 1.0	1.6 ± 0.5	1.4 ± 0.4	1.3 ± 0.3	1.3 ± 0.2	1.3 ± 0.2	0.023	1.5 ± 0.4
Post-Qp/Qs	1.0 ± 0.8	1.1 ± 0.2	1.0 ± 0.1	1.0 ± 0.1	1.0 ± 0.1	1.0 ± 0.0	0.716	1.0 ± 0.0

Ao = aorta; MPA = main pulmonary artery; AoA = aortic ampulla of PDA; NOC = Nit-Occlud^®^ coil; PDA = patent ductus arteriosus; PuA = pulmonary ampulla of PDA; PI = PDA isthmus; DD = distal diameter of NOC; PD = proximal diameter of NOC; Qp/Qs = pulmonary blood flow/systemic blood flow; Pre- = before PDA occlusion using NOC; Post- = after PDA occlusion using NOC; Pr = pressure.

**Table 2 jcm-11-02469-t002:** Characteristics of PDA occlusion using NOC according to the PDA size groups.

	Size Groups of PDA Isthmus (PI, mm) at Catheterization			
	Small (0.5 ≤ Sizes < 2)	Moderate (2 ≤ Sizes < 4)	Large (4 ≤ Sizes ≤ 9)	*p* Value
Numbers (*n* = 361)	101	210	50	
Age, m	28 ± 24	59 ± 106	202 ± 220	<0.001
Body weight, kg	13 ± 7	19 ± 16	38 ± 21	<0.001
Pre-Ao Pr, mmHg				
Systole	109 ± 19	112 ± 18	123 ± 22	<0.001
Diastole	64 ± 15	62 ± 13	67 ± 13	0.191
Mean	82 ± 17	83 ± 15	88 ± 13	0.111
Pulse Pr	46 ± 11	50 ± 12	56 ± 19	<0.001
Pre-MPA Pr				
Systole	27 ± 7	29 ± 8	32 ± 11	0.010
Diastole	11 ± 4	13 ± 6	14 ± 7	0.004
Mean	18 ± 5	20 ± 7	21 ± 9	0.009
Pulse Pr	16 ± 6	17 ± 6	18 ± 7	0.284
Pr gaps of Pre-Ao and pre-MPA				
Systole	81 ± 19	83 ± 16	94 ± 20	<0.001
Diastole	52 ± 15	49 ± 13	54 ± 13	0.055
Mean	64 ± 17	63 ± 13	69 ± 12	0.069
Post-Ao Pr				
Systole	117 ± 12	113 ± 16	124 ± 18	0.197
Diastole	63 ± 6	66 ± 11	73 ± 12	0.167
Mean	88 ± 6	85 ± 11	91 ± 12	0.270
Pulse Pr	53 ± 15	47 ± 13	51 ± 15	0.656
PDA sizes, mm				
AoA	5.8 ± 2.7	8.5 ± 3.6	13.2 ± 5.2	<0.001
PuA	4.4 ± 1.5	5.2 ± 1.6	7.2 ± 2	<0.001
Length	7.8 ± 3.1	9.9 ± 3.7	13.2 ± 5.2	<0.001
PI	1.2 ± 0.3	2.6 ± 0.5	4.6 ± 1.1	<0.001
Size gaps of PDA and NOC, mm				
AoA—DD	0.9 ± 2.5	2 ± 3	4 ± 6	<0.001
PuA—PD	−0.3 ± 1.5	−0.4 ± 1.7	0.6 ± 1.8	0.130
PI—DD	−3 ± 0.9	−4 ± 2	−4 ± 2	0.201
PI—PD	−3 ± 0.6	−3 ± 1.0	−1 ± 1	<0.001
Pre-Qp/Qs *	1.4 ± 0.4	1.4 ± 0.4	1.5 ± 0.4	<0.001
Post-Qp/Qs *	1.0 ± 0.1	1.0 ± 0.9	1.0 ± 0.0	0.658
PDA types A/B/C/D/E/U, *n*	45/6/5/6/34/5	130/16/11/4/24/25	37/4/5/0/4/0	
RS, *n*				Total RS/COc
Immediate time	12 (12%)	65 (31%)	13 (26%)	90 (25%)/271 (75%)
1 day	11 (11%)	53 (25%)	10 (20%)	74 (20%)/287 (80%)
1 month	4 (4%)	26 (12%)	5 (10%)	35 (10%)/326 (90%)
6 months	3 (3%)	8 (4%)	4 (8%)	15 (4%)/346 (96%)
1 year	1 (1%)	4 (2%)	3 (6%)	8 (2%)/353 (98%)

* The *p* values between pre-Qp/Qs and post-Qp/Qs were < 0.001 at small-, moderate-, and large-sized PDA groups, respectively. Ao = aorta; MPA = main pulmonary artery; AoA = aortic ampulla of PDA; COc = complete occlusion; NOC = Nit-Occlud^®^ coil; PDA = patent ductus arteriosus; PuA = pulmonary ampulla of PDA; PI = PDA isthmus; DD = distal diameter of NOC; PD = proximal diameter of NOC; Qp/Qs = pulmonary blood flow/systemic blood flow; Pre- = before PDA occlusion using NOC; Post- = after PDA occlusion using NOC; Pr = pressure; RS = residual shunt.

**Table 3 jcm-11-02469-t003:** Morphological types and numbers (*n*) of PDA subjects according to the chronological age groups.

	Pediatric Ages						Adults	Total Ages
	≤6 Months	7–12 Months	1–6 Years	7–12 Years	13–18 Years	Subtotal	Subtotal	
PDA Types	*n* = 26	*n* = 57	*n* = 175	*n* = 57	*n* = 10	*n* = 325	*n* = 36	*n* = 361
A (conical)	16 (62%)	35 (60%)	115 (65%)	16 (28%)	6 (60%)	188 (58%)	21 (58%)	209 (58%)
B (window)	1 (4%)	3 (4%)	12 (7%)	4 (7%)	1 (10%)	21 (6%)	5 (14%)	26 (7%)
C (tubular)	1 (4%)	2 (4%)	14 (8%)	1 (2%)	1 (10%)	19 (6%)	2 (6%)	21 (6%)
D (complex)	0 (0%)	2 (4%)	5 (3%)	1 (2%)	2 (20%)	10 (3%)	0 (0%)	10 (3%)
E (elongated)	6 (23%)	13 (21%)	29 (17%)	8 (14%)	0	56 (17%)	2 (6%)	58 (16%)
U (unclassified)	2 (7%)	2 (7%)	0	27 (47%)	0	31 (10%)	6 (16%)	37 (10%)

PDA = patent ductus arteriosus.

**Table 4 jcm-11-02469-t004:** The pressure differences before and after PDA occlusion using NOC in aorta.

	Pediatrics		*p*	Adults		*p*
	Before	After	Wilcoxon Test	Before	After	Wilcoxon Test
Systole Ao Pr	111 ± 17	114 ± 15	0.031	129 ± 26	140 ± 17	0.317
Diastole Ao Pr	63 ± 14	67 ± 12	0.130	66 ± 11	73 ± 3	0.421
Mean Ao Pr	83 ± 15	86 ± 11	0.535	89 ± 11	90 ± 10	0.125
Pulse Pr of Ao	48 ± 10	47 ± 13	0.004	67 ± 14	64 ± 25	0.321
Qp/Qs	1.4 ± 1.0	1.0 ± 0.8	<0.001	1.5 ± 0.4	1.0 ± 0.0	<0.001

Ao = aorta; NOC = Nit-Occlud^®^ coil; PDA = patent ductus arteriosus; Pr = pressure (mmHg); Qp/Qs = pulmonary blood flow/systemic blood flow.

**Table 5 jcm-11-02469-t005:** Various comorbidities of congenital heart disease (CHD) accompanying PDA occlusion using NOC.

The CHDs Accompanying PDA	Numbers (*n* = 361)
Simple patent ductus arteriosus	329
Secundum atrial septal defect	11
Ventricular septal defect	8
Double outlet right ventricle with sub-pulmonary ventricular septal defect (Taussig-Bing anomaly)	3
Endocardial cushion defect	3
Pulmonary atresia with intact ventricular septum	2
Total anomalous pulmonary venous return, pulmonary atresia with complete endocardial cushion defect	1
Ebstein anomaly	1
Partial fusion of right coronary cusp and non-coronary cusp without significant aortic stenosis	1
Pulmonary valve stenosis	1
Hypertrophic cardiomyopathy	1

NOC = Nit-Occlud^®^ coil; PDA = patent ductus arteriosus.

**Table 6 jcm-11-02469-t006:** Complications with PDA occlusion using NOC at 1 year after the procedure.

	Numbers	Small-Sized PDAs	Moderate-Sized PDAs	Large-Sized PDAs
Residual shunts	8			
Persistent residual shunts	6	4	2	
Double NOCs to occlude residual shunts	2			2 (7/5 + 4/3, 9/6 + 7/6)
Flow disturbances in the PA by NOC	7			
Mild narrowing	5	3	2	
Stenosis of PA (particularly left PA)	2		2	
Flow disturbances in the aorta by NOC				
Mild narrowing	2	1	1	
Atrial fibrillation	1			1 (42 years old)
Total events	18/361 = 5%	8/361 = 2%	7/361 = 2%	3/361 = 1%

NOC = Nit-Occlud^®^ coil; PDA = patent ductus arteriosus; PA = pulmonary artery.

## Data Availability

The data underlying this article will be shared at reasonable request to the corresponding authors.

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
