# Peer review of "Safety and Efficacy of the Nit-Occlud® Coil for Percutaneous Closure of Various Sizes of PDA"

_jcm, 2022, doi:10.3390/jcm11092469_

Round 1

Reviewer 1 Report

The authors reported their experience with Nit-Occlud® coil for percutaneous closure of various sizes of PDA. This paper shows that PDA occlusion using NOC can be a safe and feasible procedure to close various sizes and types of PDA without complications : PDA occlusion using 25 NOC is as effective and safe as ADO for the occlusion of PDAs of all sizes.

The paper is very interesting and well structured. Statistical analysis and study design seems to be correct and well documented.

There are only few grammar/spelling checks to perform.

Author Response

We appreciate your acknowledgement of our study. We conducted English language editing through company Editage (www.editage.co.kr).

Reviewer 2 Report

Interesting work and paper.

I would suggest two things: 1st define in methods measurements and ratios that are used in Results; and explain the device in Methods (size and choice of device size per PDA size) and very especially the various "gaps" it is referred to in the Results (those gaps are not known in the literature, especially not in this nomenclature, they deserve clarification).

For instance, the "gaps between PDA and NOC", which diameter of teh NOC was taken into account?

In terms of actual results, I think that the proportion of residual shunt is OK, but could do better, and must be acknowledged so.  It is worst in larger PDA as expected.

I am not sure about the importance of comparaison between the pediatric and adult groups!!! It is cumbersome for no actual interest or value.  A graphic depiction of gradual increase or decrease in size/gaps from early infancy to adulthood would provide a clearer picture (to support Discussion).

PDA type in adult was A > B > E; in children it was A > E, with the E type being decrescendo as age groups progressed. Authors are invited to report prevalence of premature birth among children's groups (at least). 

No othjer comments except that data may be simplified using useful graphs to get the main message accross.

Author Response

We appreciate your acknowledgement of our study. Your comments have encouraged us to improve this study and add more valuable information. Thank you very much. Please see the attachment. 
